# Toxic Metal Implications on Agricultural Soils, Plants, Animals, Aquatic life and Human Health

**DOI:** 10.3390/ijerph17072204

**Published:** 2020-03-25

**Authors:** Uchenna Okereafor, Mamookho Makhatha, Lukhanyo Mekuto, Nkemdinma Uche-Okereafor, Tendani Sebola, Vuyo Mavumengwana

**Affiliations:** 1Department of Metallurgy, School of Mining, Metallurgy and Chemical Engineering, Faculty of Engineering and the Built Environment, University of Johannesburg, Auckland Park 2006, South Africa; emakhatha@uj.ac.za; 2Department of Chemical Engineering, School of Mining, Metallurgy and Chemical Engineering, Faculty of Engineering and the Built Environment, University of Johannesburg, Auckland Park 2006, South Africa; lukhanyom@uj.ac.za; 3Department of Biotechnology & Food Technology, Faculty of Science, University of Johannesburg, Auckland Park 2006, South Africa; nucheokereafor@gmail.com (N.U.-O.); tendanisebola@rocketmail.com (T.S.); 4South African Medical Research Council Centre for Tuberculosis Research, Division of Molecular Biology and Human Genetics, Department of Medicine and Health Sciences, Stellenbosch University, Stellenbosch 7600, South Africa; vuyom@sun.ac.za

**Keywords:** toxic metals, contamination, toxicity, human health

## Abstract

The problem of environmental pollution is a global concern as it affects the entire ecosystem. There is a cyclic revolution of pollutants from industrial waste or anthropogenic sources into the environment, farmlands, plants, livestock and subsequently humans through the food chain. Most of the toxic metal cases in Africa and other developing nations are a result of industrialization coupled with poor effluent disposal and management. Due to widespread mining activities in South Africa, pollution is a common site with devastating consequences on the health of animals and humans likewise. In recent years, talks on toxic metal pollution had taken center stage in most scientific symposiums as a serious health concern. Very high levels of toxic metals have been reported in most parts of South African soils, plants, animals and water bodies due to pollution. Toxic metals such as Zinc (Zn), Lead (Pb), Aluminium (Al), Cadmium (Cd), Nickel (Ni), Iron (Fe), Manganese (Mn) and Arsenic (As) are major mining effluents from tailings which contaminate both the surface and underground water, soil and food, thus affecting biological function, endocrine systems and growth. Environmental toxicity in livestock is traceable to pesticides, agrochemicals and toxic metals. In this review, concerted efforts were made to condense the information contained in literature regarding toxic metal pollution and its implications in soil, water, plants, animals, marine life and human health.

## 1. Introduction

The overwhelming discharge of harmful contaminants into the environment as a result of escalated urbanization and industrialization remains an issue of interest in an era of reported climate change. There is a widespread ecological and global public health concern related to the contamination of the ecosystem on the account of toxic metals. Humans have been further exposed to these toxic metals that emanate from various industrial, agricultural, domestic and technological processes [1]. Renowned sources of toxic metals in the environment include agricultural, pharmaceutical, atmospheric, industrial, domestic effluents and geogenic activities [2]. Environmental degradation is a common occurrence in point source areas such as mining, foundries and smelters, and other metal-based industrial operations [3,4].

Despite the natural occurrence of these toxic elements within the earth’s crust, anthropogenic activities such as mining, electroplating, smelting operations, domestic and agro-allied industries, and geogenic activities are responsible for environmental contamination and human exposure to toxic metals [2]. Natural phenomena such as weathering and volcanic eruptions have been observed to be sources of toxic metal pollution [3,4], while metal processing in refineries, coal burning in power plants, petroleum combustion, nuclear power stations, microelectronics, wood preservation and paper processing plants are some of the important industrial sources [1,2]. Environmental degradation is a common occurrence in point source areas such as mining, foundries and smelters, and other metal-based industries [3,4]. So far, it has been demonstrated that toxic metals in the environment have multiple sources, but the purpose of this review is on mine tailings. 

The economies of countries with intense anthropogenic activities such as mining are reported to be favourable; however, there are devastating implications to the environment and human health in the long run [5]. Mining-related operations generate excessive waste materials in the forms of debris and tailings, which are subsequently released into the environment [6]. These wastes are composed of both harmful and toxic metals with certain noble metals. Toxic metal is usually a generic term used to describe any natural occurring metal that could neither be degraded nor destroyed but specifically areknown to be toxic to humans, and these include antimony, arsenic, beryllium, bismuth, cadmium, lead, mercury and nickel [7]. Due to inadequate skills, technology and poor management, the ecosystem becomes the receptor of these waste materials. 

Contaminated waters by acid mine drainage results in colour formation and turbidity due to the high concentration of trace metals. Nevertheless, there are instances where contamination is not easily detectable to the naked eyes, thus making it practically impossible to notice its occurrence. Toxic metals such as Cu, Zn, Co, Ni, Fe, Cr, Mn, I and Se widely known as micronutrients play a crucial role in the metabolic and physiological activities of humans, plants as well as microorganisms depending on their concentrations. On the other hand, specific toxic metals such as As, Ag, Hg, Cd and Pb are of no biological relevance to plants and animals; rather. they are harmful. At higher concentration levels, these elements pollute the environment, resulting in deleterious health implications for humans, plants and animals [8,9,10]. Being noxious and containing carcinogenic metalloids, toxic metals could result in cancer of the skin and lungs and in urinary tract disorders, cardiovascular diseases, neurotoxicity as well as diabetes [11,12,13].

Based on the above, this review article seeks to address some of the environmental health challenges in places with previous and ongoing mining activities in a bid to ensure better practices in current mine processes to safeguard our environment. In addition, this review seeks to bring public awareness on the effects of tailings on human lives and the environment such that legislative bodies can be pressured to develop stricter legislative frameworks on tailing management.

## 2. The Description of Mine Tailings and Tailings Dam

The materials that are left over in the form of liquids and solid or a slurry of very fine particles upon successful separation and processing of minerals (elements) of interest from an ore are collectively termed mine tailings [14]. Often, tailings are of no financial benefits to mining firms. They are composed of fine particles suspended in water, with the potential of causing havoc to the environment through the release of trace elements, causing surface runoffs and sinkholes and contamination of the environment. These materials are very distinct from the usual mine overburden, which includes the soil and rock that are removed to gain access to the ore deposits in open pit mines [15,16] and waste rocks, which are rocks that are mined but contain minerals in very low concentrations to be extracted at a profit and are, therefore, removed ahead of processing [17].

In mining, the steps taken for product extraction are not always efficient, thus creating challenges in recovering reusable and expended processing reagents and chemicals. Most of the unrecoverable and uneconomic metals, chemicals and process water are disposed, largely as slurries, to a final storage area known as a Tailings dam (TD). 

In most cases, tailings are pumped at very high pressure into ponds for sedimentation to occur. As a cost-saving venture, the water ejected from such mine tailings dam are often employed in other processing cycles at the mines. Mine tailings are regularly stored in tailing dam. Despite not having the exact number of global tailing dams, when poorly designed, constructed or managed, they pose a significant risk to local communities and ecosystems, particularly in downstream environments. The construction of such retaining structures is exclusive and dependent on the type of environment and mineral processing operation.

## 3. The Production of Mine Tailings

The disposal of mining wastes is one of the sources of environmental impact for many mining exploration activities due to the generated volume of wastes exceeding the actual in situ total volume of processed ore [18]. In recent times, there have been the continued increase in this trend following the demand for more minerals and metals [19]. Currently, the volume of tailings being produced by industries are at an alarming rate of over 173.64 billion tons. The production of these tailings and the challenges associated with their storage could be linked to the mineral processing techniques being adopted by individual mines.

Run of mine (ROM) ores get reduced physically through processes such as crushing and grinding. However, the extent of grinding is solely dependent on the metallurgical methods applied in the removal of the economic product. A suitable extraction method is based on a series of mineralogical investigation which could reveal other minerals of economic relevance. In addition, such investigations reveal the quantities and type of reagents that are best suited in separating the concentrates (target elements/minerals) from the gangue (unwanted) materials and the appropriate storage methods for the tailings [20]. Processing reagents and characteristics of mine tailings such as particle size may be determined using several plant tests. Although, these series of tests may not be a true representation of the final tailings generated from the operational full-scale plant. This implies that there is a provisional design of tailing dams which then gets confirmed when there is an actual production of tailings [21].

Concentration is a metallurgical process of extracting the economic product from the crushed and ground ore, leaving behind unwanted materials or wastes often referred to as tailings [22]. As the first step in the mineral processing sequence, Froth flotation involving the use of chemical reagents is a common concentration technique while gravity and magnetic separation are other methods utilized [18]. In gold processing, for instance, gravity separation is employed in recovering the coarser particles, while leaching is applied for the finer fractions [23].

## 4. Characteristics of Mine Tailings

Tailing characteristics are unique and may be attributed to several factors. Often, sediments from mine tailings portray physical and chemical properties that are like specific river sand and silt. The determination of the actual properties of tailings are based on geochemistry, nature and mineralogy of the ore coupled with the processes utilized in extraction of various economic products [24,25]. There have been reported cases of tailings possessing different mineralogy despite being generated from the same source [20]. A gold mine tailing, for instance, exhibits weak aggregation; limited cohesion potential, which results in different moisture levels and temperatures; high hydraulic conductivity; and fine texture, which makes them different from the soil [26,27]. From a chemical standpoint, gold tailings have high salinity and are composed of 6% pyrite with very little organic matter [28]. High acidity together with high metal levels in ground water within the proximity of gold tailings is due to high sulphide content [26]. In terms of pH values, tailings from Iran were reported to be 7.35 [29] and 3.25–6.28 in South Africa [30,31] while, in India, it was 3.48–8.12 [32].

In order to ascertain the likely dangers associated with tailings when deposited in a storage facility, a consideration of its characteristics cannot be overlooked. Upon the determination of the distinct features of the generated tailings, an appropriate design requirement may be recommended to alleviate environmental effect and ideal operational routine. 

The water balance of a mining project from a design perspective is influenced by the liberation of water by the tailings upon discharge into a storage facility and the subsequent volume that is available for return pumping to the processing plant. The liberation potential is dependent on the physical properties of the tailings, which may be estimated based on different laboratory tests. 

As a prerequisite to determining the design requirements of a mine tailing storage facility, certain properties such as chemical composition, physical composition and stability, behaviour under pressure and consolidation rates, erosion stability, settling, drying time and densification behaviour after deposition of the tailings need to be established [23]. In most cases, the degree of thickening coupled with the method of deposition influences the engineering characteristics of tailings. It is therefore pertinent to ensure that, while investigating the properties of tailings, physical features and material parameters such as beach slope angles, particle size segregation and water recovery that can occur as a result of varied deposition techniques are identified [33].

## 5. Management of Mine Tailings

Mine tailings are often stored on the surface (within retaining structures or in dry stacks) or underground (voids) by a process known as backfill which provides ground and wall support, improves ventilation, substitutes for surface tailings storage and prevents subsidence [23]. 

The problems emanating from tailing storage facilities are on the rise. Recent technological developments encourage the exploitation of lower grade ores, thus resulting in the generation of higher volumes of tailings that require safe storage. The continued review of global environmental regulations is creating stiffer requirements for various stakeholders within the mining industry on best tailing storage practices. Many historical tailing-related incidents could be attributed to poor day-to-day management, which has led to the strengthening of regulations controlling today’s tailing storage facilities [23]. The parameters that influence stability, operation and management of tailing facilities have been identified and presented together with their methods of control, intervention and mitigation. A free novel online database called TailPro (www.tailpro.com) has been developed to assist tailing personnel in the implementation of a tailing management system more efficiently and effectively.

Certain factors which impact the site selection, storage and tailing discharge methods adopted are considered in the design of a tailing storage facility [20]. The environment and ground conditions constitute major parameters that control tailing storage methodology which imperatively affects the way a facility is designed, built, operated and closed. On account of this, a range of other methods of tailings storage and discharge techniques need to be considered when designing a facility for a location. In industry, this is achieved by implementing a trade-off study, usually during the pre-feasibility stage of project development. A selection of options from this study can be taken through to the feasibility stage to assess environmental, social, economic and associated risk and operational factors with a higher level of confidence.

## 6. Ecotoxicity

Many pollutants (mining wastes, chemical and organic fertilizers, pesticides, industrial wastes and other materials) are held in soils that often contribute to water and air pollution. Since soil is a key component of environmental chemical cycles, the quality of soil and climate of society are priceless assets, as they determine the level of agricultural productivity [34,35].

Ecotoxicological tests are complementary tools used to chemically analyze soil contamination. Assessment of the behaviour and toxicity of soil elements, or compounds, should not be based solely on chemical indices. The inclusion of biological indicators in such investigations will aid in the provision of a better understanding of the behaviour of chemicals in the environment [36]. Terrestrial ecotoxicology attempts to reveal some of the deleterious, morphological, behavioural, physiological, biochemical and cytogenetic consequences of the discharge into the environment of potentially toxic elements on organisms [36,37].

Organisms in soil fauna known as bioindicators such as Collembola, earthworms, nematodes and enchytraeids are often used to indicate environmental changes at their early stages and to identify several modification types before such changes become drastic besides determining the pollution types capable of affecting a given ecosystem.

In addition, these organisms play crucial roles in monitoring and making more accurate and less impacting decisions on soil management. The use of bioindicators in monitoring programs helps detect environmental changes at their early stages or the effectiveness of measures taken to improve environmental quality [37].

## 7. Metal Toxicity

Most metabolic and physiological processes in plants and animals (humans and microorganisms) are influenced by toxic metals [38]. Some potentially toxic metals such as Cu, Co, Zn, Ni and Cr serves as both micronutrient and vital ingredients in redox processes. These metals through osmotic pressure regulation, electrostatic interactions and cofactor support the stabilization of molecules for many enzymes and, thus, the essential role of toxic metals in intricate biochemical processes [39]. Ag, As, Cd, Pb and Hg which are nonessential toxic metals are of little or no significant biological relevance to living creatures and rather are highly noxious when noticed in the environment. Studies using culture dependent and independent approaches revealed high concentration levels of toxic metals in mining effluents which impacts the diversity, the population size and the whole activity of bacteria [40,41,42]. 

Metal toxicity is a widely reported environmental health problem that is dangerous due to bioaccumulation via the food chain which could result in hazardous implications in humans and animals [43,44]. The hazardous effects of toxic trace metals (elements) are dependent on certain factors such as the dietary concentration of the elements, absorption of such elements by the system, homeostatic control of the body for such elements and the species of the animal involved [45]. In addition, the oxidation state of a toxic metal plays a role in the toxicological and biological effects to the environment. Metal toxicity could be attributed to changes in the conformational structure of nucleic acids and proteins or by interference with oxidative phosphorylation and osmotic balance [46]. In recent times, due to technological, industrial and agricultural advancement, toxic metal pollution has emerged a severe health problem. Toxic metals from industrial and electronic wastes contaminate the entire environment. Cd, Pb, Zn and Hg are some of the toxic metals of deleterious effects to humans and animals alike. 

## 8. Potentially Toxic Metals in Soil and Their Implications

Several studies into soil contaminants from anthropogenic activities in different geographical locations had been undertaken in which the findings revealed alarming levels of soil micronutrients and macronutrients which are said to be Chemical Time Bombs (CTBs) [47]. The quality of a soil contributes to both enzymatic and microbial activities [48,49]. The extent of soil contamination could be evaluated using microbial biomass as an important indicator [50]. There is a significant inhibition of microbial activity in any soil with the problem of toxic metal contamination. In a study of soil contaminated by Cu, Zn, Pb and other toxic metals, a lower microbial biomass was observed within soil nearer to a mine compared to those that are far away from the mine [51]. The relationship between the level of toxic metals and how they impact soil microbial biomass have been studied and reveals that low levels of toxic metals support microbial growth which increase microbial biomass while high concentrations could decrease soil microbial biomass expressively [52,53]. This trend is likewise applicable to enzymatical activities of soil.

The presence of mine tailings in soil leads to acidification. This is because the toxic metal ions are normally contained in untreated mine tailings. Low pH tailings, i.e. acidic tailings, contain higher amounts of toxic metals as compared to high pH tailings. At high pH, most of the toxic metal ions form insoluble hydroxides and sulphides, which then precipitate to reduce the ion content of the mine tailings [54,55,56,57,58]. Toxic metal such as Cr (III) that is characterized by a high ionic charge is most likely to be adsorbed on soil exchange sites to the soils with a relatively high cation exchange capacity (CEC). Among the widely known toxic metals, Cu (II) and Pb (II) have a greater tendency (after Cr) to be specifically adsorbed on soil and separated out from Cd, Ni and Zn [59].

In addition, the absorption of soil toxic metals by plants is not always a problem in the interim but becomes something to worry about in the long term. This is when the concentrations of these toxic metals become too high and have exceeded the permissible limit, which result in plant poisoning and subsequent death. Researchers in Florida, USA, observed that citrus seedlings were severely affected in soil with copper content of more than 50 mg/ kg, while the withering of wheat occurred at an increased level of 200 mg/ kg [60]. In a similar fashion, there was retarded growth and development patterns in the seedlings of bean and cabbage at Cd concentration of 30 μ mol/L [61]. There are reports indicating that Cd in soil may result in poor photosynthesis and protein synthesis in crops, thus damaging cell membranes [62,63]. Higher concentrations of Zn in soil suppresses plant metabolic activities, thus resulting in stunted growth and senescence.

The health of humans suffers a great setback when soils have excessive levels of toxic metals. This is attributable to absorption of toxic metals via the skin, dust inhalation and the pollution of food, water and air that constitutes the food chain. In a test conducted in China on the level of Pb in the blood of children, it was found that over 30% of the sampled cases had Pb that exceeded the standard home requirement (100 g/L), which was linked with the soil dusts [64]. The Agency for Toxic Substances Management Committee (ATSMC) outlined Cd as the world’s sixth most harmful substance that destroys human health [65]. The metabolism of Calcium is interrupted by Cd, which causes calcium deficiency, thus resulting in bone fractures and cartilage diseases.

## 9. Potentially Toxic Metals in Water and Their Implications

Globally, water contamination from toxic metals has remained a nightmare following the increasing deaths of farm animals and humans alike as a result of diseases linked to impure drinking water. Efforts by various environmental and enforcement bodies in effectively controlling the activities that act as sources of these metals have been fruitless [66]. Most biological activities have been seriously impeded as a result of these nonessential metals. Generally, the quality of water is compromised due to the presence of toxic metals which are toxic at very high concentrations, thus impacting adversely on the health of humans, animals and plants. 

In South Africa, water is a scarce commodity with over 70% of what is being provided by the government for usage in both rural and urban areas emanating from sources like rivers, streams, lakes, ponds and springs [67]. In recent times, the Environmental Mining Council of British Columbia (EMCB) proposed a concerted effort towards the safety of the purity and quantity of water against reckless mineral exploration which could compromise the overall quality of water via increased pollution and sedimentation loads, resulting in poor water quantity which conforms to the principal of sustainable development [17,68]. 

In studies conducted on untreated sewage in Musi river, Hyderabad, India, severe contamination levels by Cd, Ni, Pb, Co, Zn and Cu with mean content of 0.025, 0.062, 0.210, 0.053, 0.003 and 0.011 ppm, respectively, were observed [69]. These values were of serious concern as they exceeded the stipulated permissible limits of WHO. A similar study carried out in South Africa investigated the possible transportation of toxic metals due to ground erosion during heavy rainfall in the Rural Mhangweni (Tzaneen, Limpopo province) and reported disturbing toxic metal concentrations such as Al (6.141 ppm), Zn (0.431 ppm), Fe (5.072), Cu (1.506), Pb (2.041) and Mn (3.918 ppm), which surpassed the maximum acceptable level of water composition as stipulated by the United States Environmental Protection Agency [7].

Toxic metal such as Hg is harmful particularly to humans when present in water that forms part of the food chain; this is due to the possibility of the central nervous system being attacked by it, thus leading to Minamata disease [70,71]. The importance of Zn as a major constituent of many enzymes involved in metabolic reactions as well as the production of hormones cannot be overlooked. However, when excessively consumed by humans, it may lead to severe abdominal pain, intense vomiting, collapse and deteriorating changes in the liver [72]. With devastating lifetime effects, lead poisoning from water had been linked to stunted growth in children, damage of the nervous system, learning disabilities and, recently, crime and antisocial behaviours [73].

## 10. Potentially Toxic Metals in Plants and Their Implications

Just like animals and other living organisms, plants show some reactions towards the availability and seldom lack of critical micronutrients due to the many roles they play in metabolic processes. When these metal ions are limited or not readily available for plant uptake, they result in deficiency in growth, whereas when in excess, Cd, Hg, As, Pb and Se are supposedly deleterious. Several factors such as the growing environment, temperature, soil pH, soil aeration, competition between plant species, the root system, the availability of the elements in the soil, the type of leaves and soil moisture are important in the uptake of metals by plants [74]. On account of the environment of cultivation, previous studies revealed that an increase in pH, i.e. the environment becoming more alkaline, with a corresponding decrease in Eh (redox potential), i.e. the environment becoming more reducing, abruptly reduce the amount of toxic metals that are available to plants [75].

The detrimental effects of excessive toxic metals towards plant growth had been widely documented [76,77,78]. Mn, Pb, Cd, Cr and Co during a study were observed to be responsible for the poor growth of maize plants (Zea mays L.) [77]. At extreme levels, toxic metals could result in oxidative stress in plants, mutilation of cell structure through the substitution of deficient elements with toxic metals and slow down photosynthetic processes in plant cells [79]. The phytotoxicity effect of Zn and Cd is seen by retarded growth and development, metabolism and an inductive oxidation damage in various plant species such as Brassica juncea [80,81]. At very high concentrations, Cd and Zn could result in oscillation in catalytic proficiency of enzymes in pea plants [82]. Zinc toxicity has been linked to restricted growth of both root and shoot in plants [83] as well as chlorosis in newer leaves [84]. There are reported cases of reduced crop production due to Ni toxicity that impaired certain enzymatic activities (amylase, protease and ribonuclease), thus adversely affecting the germination of seeds. Also affected by Ni were activities such as membrane stability, nitrate reductase and carbonic anhydrase [85]. 

As a micronutrient for plants, Cu is crucial in the synthesis of ATP and assimilation of CO_2_ [86]. Cu constitutes a major part of proteins such as plastocyanin of photosynthetic system and cytochrome oxidase of respiratory electron transport chain [87]. When in excess as a result of anthropogenic activities such as mining, Cu has a cytotoxic effect on soil which induces stress and causes growth retardation and leaf chlorosis in plants [88]. Oxidative stress in plants as a result of high levels of Cu results in the disturbance of metabolic pathways and damage to macromolecules [89]. From previous studies, Copper toxicity was reported to have affected the growth of *Alyssum montanum* [90] while a combination of both Cu and Cd were responsible for the poor germination, seedling length and number of lateral roots in *Solanum melongena* [91].

Occurring in several forms, such as HgS, Hg^2+^, Hg^0^ and methyl-Hg, Mercury (Hg) is known to accumulate in higher and aquatic plants [92,93,94]. It is reported that higher concentrations of Hg^2+^ is strongly phytotoxic to plant cells, inducing visible injuries and physiological disorders in plants such as the closure of leaf stomata and physical obstruction of the flow of water [95,96]. There are observed interferences in the mitochondrial activity of plants as a result of high levels of Hg^2+^, which results in the disruption of biomembrane lipids and cellular metabolism [97,98].

Lead (Pb) exerts detrimental effects on the morphology, growth and photosynthetic processes of plants through interference with vital enzymes, thus inhibiting seed germination [99,100]. Oxidative stress is another complication as a result of higher concentrations of Pb which increases the production of reactive oxygen species (ROS) in plants [101].

It is noteworthy to mention that there are plants known as accumulators that can withstand higher concentrations of heavy metals in their natural environment. These plants are able to tolerate high metal levels through diverse mechanisms such as (i) exclusion: restriction of metal transport and maintenance of a constant metal concentration in the shoot over a wide range of soil concentrations; (ii) inclusion: metal concentrations in the shoot reflecting those in the soil solution through a linear relationship; and (iii) bioaccumulation: the accumulation of metals in the shoot and roots of plants at both low and high soil concentrations [102]. A summary of some of the effects of metal toxicity on plants are detailed in Table 1. 

## 11. Potentially Toxic Metals in Human Health and Their Implications

A wider population in most developing countries is faced with challenges of toxic metal contamination of dietary substances due to poor legislations on the management of toxic metal sources coupled with plant uptake of metals at high concentrations. [136,137,138]. On account of these metals being ubiquitous and recalcitrant, their admission into the human body poses severe health implications as they could result in the malfunctioning of certain cellular processes through the displacement of essential metals from their respective locations. Toxic metals such as Lead (Pb), cadmium (Cd), mercury (Hg) and arsenic (As) are extensively dispersed in the environment. Unfortunately, there are no known benefits of these metals in humans as well as any established homeostasis mechanism for them [139]. In addition to the toxicities of metals, the potential carcinogenicity of metal compounds had been of interest to society. With a growing working population in the mining industries coupled with human settlements springing forth within the vicinities of mines, the health conditions of individuals in this circumstance are compromised due to continued exposure to various trace metals.

The problem of oxidative deterioration of biological macromolecules has been linked with the binding of metals to DNA and nuclear proteins [140]. Some of the symptoms of metal poisoning in humans within the mining industries include intellectual disability in children, dementia in adults, central nervous system disorders, kidney diseases, liver diseases, insomnia, emotional instability, depression and vision disturbances [141]. 

The transportation mechanism of toxic metals in humans is somewhat complex. For instance, Pb ends up in humans mostly via the digestive tract and respiratory tract, before going into the blood circulation in the form of soluble salts, protein complexes or ions with over 95% of the insoluble phosphate lead accumulating in bones. As a highly pro-organizational element, Pb affects and destroys several body organs and systems, such as kidney, liver, reproductive system, nervous system, urinary system, immune system and the basic physiological processes of cells and gene expression [142]. Prolong lead exposure in children has been linked with neurological damage resulting to a decrease in intelligence, loss of short-term memory, learning disabilities and challenges with general coordination. Prenatal exposure may result in the reduction in immunity and birth weight, which alludes to the claims for why some infants are diagnosed with asthma and allergies [143]. There are suggestions that lead can impact behavioural inhibition mechanisms with a resultant intensification in violence [144] and that it can support tooth decay [145].

Cu, Zn and Ni, on the other hand, are vital trace metals in the human body but, when consumed in excess, are deleterious to the body. With a larger population working in the mining industries, the carcinogenic effects of Ni and Cu, both being regarded as tumor promoting metals, have raised international concerns. Prolonged human exposure to copper often results in severe mucosal irritation and corrosion, capillary damage, hepatic and renal damage, and irritation of the central nervous system occasioned with depression. System dysfunctions resulting in the impairment of growth and reproduction are linked with excessive zinc. A previous study indicated that individuals working closely with nickel powder are at risks of having respiratory cancer and that the content of Ni in the environment is absolutely associated with nasopharyngeal carcinoma [146]. With symptoms such as uncontrollable shaking, muscle wasting, partial blindness and deformities in children exposed in the womb, excessive mercury damages the nervous system [147]. There are reports suggesting that, at concentrations lower than the stipulated limits of WHO, mercury can damage both the foetal and embryonic nervous systems with consequential learning complications, poor memory and shortened attention spans [148].

Depending on the severity of the level of exposure, Cadmium (Cd) toxicity is mostly evident in organs such as liver, kidneys, placenta, brain, lungs and bones [149]. Some of the symptoms of the deleterious effects include nausea, abdominal cramps, vomiting, dyspnea and muscular weakness. Extreme exposure has been linked to death and pulmonary odema with effects such as emphysema, bronchiolitis and alveolitis [150]. A variety of clinical conditions such as cardiac failure cancers, anosmia, osteoporosis, cerebrovascular infarction, proteinuria cataract formation in the eyes and emphysema are largely associated with cadmium.

Just like mercury and lead, the toxicity symptoms of arsenic are dependent on the form in which they are ingested. Arsenic aids the coagulation of protein, aid the formation of complexes with coenzymes and stops the production of adenosine triphosphate (ATP) during respiration. It is mostly carcinogenic in compounds of its oxidation states. which results in death when exposed to an extreme level. Cases of arsenic are responsible for disorders similar to and often likened to Guillain-Barre syndrome, an anti-immune disorder that arises when the body’s immune system erroneously attacks part of the PNS, leading to inflammation of the nerve that causes weakness of the muscle [150]. A summary of some of the effects of potentially toxic metal on human health are detailed in Table 2.

## 12. Potentially Toxic Metals in Aquatic Environment and Their Implications

As highly persistent and toxic in trace amounts, toxic metals could potentially propel aquatic animals’ acute oxidative stress. Through anthropogenic activities’ pollutants in the form of pesticides, pharmaceuticals and toxic metals contaminate several water bodies. Of these pollutants, toxic metals are of great danger to the ecology of water bodies as they influence fish, which is a vital protein source and, thus, the ecotoxicological significance of these toxic metal contaminants. Since metals are not bacterial degradable, their presence as contaminants in rivers may impact adversely the ecological equilibrium of aquatic environment, resulting in reduced diversity of marine life [197,198].

Histopathological alterations such as interference in the metabolic activities of fish resulting to cellular intoxication and death at a cellular level are brought about by toxic metals. Histological and histopathological changes in critical organs and tissues due to toxic metal pollutants often occur before they produce irreversible effects on the biota. The continuous exposure of fishes to waterborne and particulate toxic metals is a result of the constant movement of water through gills and through food sources.

Toxic metals have been reported to be responsible for the generation of Reactive Oxygen Species (ROS) which destroys the protein, lipid and DNA content of exposed aquatic animals. Redox active toxic metals (Fe, Cu, Cr, etc.) undergo redox cycling, while redox inactive toxic metals (such as Pb, Cd and Hg) undergo covalent electron sharing with cells’ major antioxidant enzymes (Thiols). Both groups of toxic metals result in the production of ROS as hydroxyl radical (OH), Superoxide radical (O_2_¯) or hydrogen peroxide (H_2_O_2_), which reduce cells’ inherent antioxidant defense [199]. From previous studies, it was reported that several defects such as epithelial lifting, interstitial oedema, leucocytic infiltration, hyperplasia of epithelial cells, lamellar fusion, vasodilatation and necrosis are a result of toxic metals coming in contact with the large surface area of fish gills [200,201,202]. The contamination of fish by toxic metals such as methylmercury, a toxic chemical form of mercury formed by bacterial methylation of organic mercury, is a public health concern as they form part of the food chain [203].

## 13. Conclusions

Anthropogenic activities such as mining and its associated metallurgical processes have contributed significantly to environmental deterioration through the improper management of the tailings which contain toxic metals that the mining industries produce. The generated tailings are one of the major metal sources which can contaminate a variety of ecological settings through particle dispersion during windy periods. During particle dispersion, animals and humans can inhale the tailing particles which contain toxic metals, thus resulting in health-related complications. Some of the devastating effects of these metals on human health include but are not limited to developmental retardation, cancer, kidney damage, endocrine disruption, immunological and neurological effects, and other disorders. Aquatic organisms suffer damages mostly at a cellular level as a result of extreme toxic metal concentrations. Considering the toxicity and bioaccumulation potentials of toxic metals, strict legislation on tailings management needs to be developed and enforced, such that the abovementioned negative effects could be curbed.

## Figures and Tables

**Table 1 ijerph-17-02204-t001:** Summary of the effects of potentially toxic metal on plants.

Toxic Metal	Plant	Toxic Effect on Plant	Reference
As	Rice (Oryza sativa)	Reduced leaf area and dry matter production; reduction in seed germination; decrease in seedling height	[103,104]
	Tomato (Lycopersicon esculentum)	Drop in fruit yield; reduction in leaf fresh weight	[105]
	Canola (Brassica napus)	Restricted growth; chlorosis; wilting	[106]
Cd	Wheat (Triticum sp.)	Decline in seed germination; reduction in nutrient content of plant	[107]
	Garlic (Allium sativum)	Reduced shoot development; Cd buildup	[108]
	Maize (Zea mays)	Reduced shoot development; inhibition of root growth	[109]
Co	Tomato (Lycopersicon esculentum)	Reduction in plant nutrient content	[110]
	Mung bean (Vigna radiata)	Decline in antioxidant enzyme actions; reduction in plant sugar, starch, amino acids and protein content	[111]
	Radish (Raphanus sativus)	Decline in shoot length, root length and total leaf area; reduction in chlorophyll content, plant nutrient content, antioxidant enzyme activities, decrease in plant sugar, amino acid and protein content	[112]
Cr	Wheat (*Triticum* sp.)	Stunted shoot and root growth	[113,114]
	Tomato (*Lycopersicon esculentum*)	Reduction in plant nutrient acquisition	[115,116]
	Onion (*Allium cepa*)	Inhibition of germination process; plant biomass reduction	[117]
Cu	Bean (*Phaseolus vulgaris*)	Buildup of Cu in plant roots; root malformation and reduction	[118]
	Black bindweed (*Polygonum convolvulus*)	Plant death; reduced biomass and seed production	[119]
	Rhodes grass (*Chloris gayana*)	Stunted root development	[120]
Hg	Rice (*Oryza sativa*)	Reduction in plant height; reduced tiller and panicle formation; reduced yield; bioaccumulation in shoot and root of seedlings	[121]
	Tomato (*Lycopersicon esculentum*)	Decrease in the percentage of germination; reduced plant height; reduction in flowering and fruit weight; chlorosis	[122]
Mn	Broad bean (*Vicia faba*)	Manganese accumulation in shoot and root; chlorosis.	[123]
	Spearmint (*Mentha spicata*)	Reduction in the content of chlorophyll and carotenoid.	[124]
	Pea (*Pisum sativum*)	Reduction in relative growth rate and photosynthetic activities.	[125]
	Tomato (*Lycopersicon esculentum*)	Slower plant growth; reduction in the concentration of chlorophyll.	[126]
Ni	Pigeon pea (*Cajanus cajan*)	Drop in chlorophyll content and stomatal conductance; decreased enzyme activity which affected Calvin cycle and CO_2_ fixation.	[127]
	Rye grass (*Lolium perenne*)	Reduction in plant nutrient acquisition; decrease in shoot yield; chlorosis.	[128]
	Wheat (*Triticum* sp.)	Reduction in acquisition of plant nutrient.	[129,130]
	Rice (*Oryza sativa*)	Stunted root development.	[131]
Pb	Maize (*Zea mays*)	Reduction in germination percentage; suppressed growth; reduced plant biomass; decrease in plant protein content.	[132]
	Portia tree (*Thespesia populnea*)	Drop in number of leaves and leaf area; reduced plant height; decrease in plant biomass.	[133]
	Oat (*Avena sativa*)	Inhibition of enzyme activity which affected CO_2_ fixation.	[78]
Zn	Cluster bean (*Cyamopsis tetragonoloba*)	Reduction in germination percentage; reduced plant height and biomass; decrease in chlorophyll, carotenoid, sugar, starch and amino acid content.	[134]
	Pea (*Pisum sativum*)	Reduction in chlorophyll content; alteration in structure of chloroplast; reduction in photosystem II activity; reduced plant growth.	[81]
	Rye grass (*Lolium perenne*)	Accumulation of Zn in plant leaves; growth reduction; decrease in plant nutrient content; reduced efficiency of photosynthetic energy conversion	[135]

**Table 2 ijerph-17-02204-t002:** Summary of the effects of potentially toxic metal on human health.

Toxic Metal	Toxic Effect on Humans	Reference
As	Cancer of the skin, lungs, bladder, prostrate and blood (Leukemia)	[151,152,153,154,155,156]
	Neurobehavioral abnormalities during puberty and adulthood	[157,158]
	Diabetes and cardiovascular disorders	[159,160]
	Increased fetal mortality and preterm birth in pregnancy	[161]
Cd	Shortness of breath, lung edema and destruction of mucous membranes	[162]
	Acute vomiting and diarrhoea	[163]
	Kidney and bone damage	[164,165]
Co	Decreased pulmonary function, increased frequency of cough, respiratory inflammation and pulmonary fibrosis	[166]
	Myelopathy, brachial plexus neuropathy and vocal cord paresis	[167]
	Hearing and visual impairment	[168]
Cr	Pulmonary irritant effects such as asthma, chronic bronchitis, chronic irritation, chronic pharyngitis, chronic rhinitis, congestion and hyperemia, polyps of the upper respiratory tract, tracheobronchitis and ulceration of the nasal mucosa with possible septal perforation	[169]
	Irritant and allergic contact dermatitis	[170,171]
	Respiratory system cancers such as lungs, nasal and sinus cancers	[172]
	Acute tubular necrosis and acute renal failure	[173]
	Derangement of liver cells, necrosis, lymphocytic and histiocytic infiltration, and increases in Kupffer cells	[174]
	Decreased haemoglobin content and hematocrit and increased total white blood cell counts, reticulocyte counts and plasma haemoglobin	[175]
Cu	Irritation of the nose, mouth and eyes; headaches; dizziness; nausea; vomiting; stomach cramps; and diarrhea	[176]
	Liver and kidney damage and even death	[177]
	Wilson’s Disease that is characterized by hepatic cirrhosis, brain damage, demyelization, renal disease and copper deposition in the cornea	[178]
Hg	Neurological and behavioural disorders such as tremors, insomnia, memory loss, neuromuscular effects, headaches, and cognitive and motor dysfunction	[179]
	Red blood cell accumulation (competes with iron for haemoglobin binding) and inhibition of myelin synthesis in developing fetus and children	[180]
	Immune, enzyme and genetic alterations	[181,182]
	Young’s syndrome (Azoospermia sinopulmonary infections)	[183]
Mn	Parkinsonian syndromes	[184]
	Alteration in cardiovascular function	[185]
	Increased infant mortality, hallucinations, forgetfulness and nerve damage	[186]
	Impotence and loss of libido in men	[187]
Ni	Nausea, vomiting, abdominal pain, diarrhea, headache, cough, shortness of breath and giddiness	[188]
	Death due to nickel-induced Adult Respiratory Distress Syndrome (ARDS), chronic bronchitis, reduced lung function, and cancer of the lung and nasal sinus	[189]
	Allergic skin reaction	[190]
	Genotoxicity haematotoxicity, teratogenicity, immunotoxicity and carcinogenicity	[191]
Pb	Headache, loss of appetite, abdominal pain, fatigue, sleeplessness, hallucinations, vertigo, renal dysfunction, hypertension, arthritis, birth defects, mental retardation, autism, psychosis, allergies, paralysis, weight loss, dyslexia, hyperactivity, muscular weakness, kidney damage, brain damage, coma and death	[192]
	Disruption of the intracellular second messenger systems resulting in the alteration of the functioning of the central nervous system	[193]
Zn	Respiratory disorder from inhalation of zinc smoke, epigastric pains, risks of prostate cancer and lethargy	[194]
	Copper deficiency	[195]
	Irritation and corrosion of the gastrointestinal tract, acute renal tubular necrosis and interstitial nephritis	[196]

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
