# Peer review of "Toxic Metal Implications on Agricultural Soils, Plants, Animals, Aquatic life and Human Health"

_ijerph, 2020, doi:10.3390/ijerph17072204_

Round 1

Reviewer 1 Report

The revised manuscript has a great improvement. The authors discussed two scientific questions. The one is mine tailings implications on environment, and the other one is toxic metals on environment and human health. It is no doubt that mining activities (including tailings, AMD, dust, etc.) are the main sources of toxic heavy metals. The authors tried to bring the two scientific problems together, but actually focus on the second scientific problem. Therefore, the title is misleading. I think it is better to revise the title as “heavy metal implications on Agricultural Soils, Plants, Animals, Aquatic life and Human Health”. Then a separate section discusses the sources of heavy metals in the environment. The impact of mining tailings can be referred to in the article (Tang Z H, Ouyang T P , Li M K , et al. Potential effects of exploiting the Yunfu pyrite mine (southern China) on soil: evidence from analyzing trace elements in surface soil[J]. Environmental Monitoring and Assessment, 2019, 191(6):395.)

Reviewer 2 Report

This study presents a review about mine tailings as a source of heavy metals and its implications on agricultural soils, plants, animals, aquatic life and human health. Due to a widespread of mining activities in South Africa, pollution is a serious health concern with devastating consequences on the health of animals and humans likewise. This paper aims to report contamination of major mining effluents from tailings. In this review, concerted efforts were made to concise the information contained in literature regarding toxic metal pollution and its implications in soil, water, plants, animals, marine life and human health.

Here are some comments about the manuscript:

  1. Article structure is not reasonable, should be divided into several parts according to the hierarchy, the same classification of content into the same category.
  2. In the “introduction” part, authors should clearly express the purpose of the research, the significance of this article should be explained.
  3. There was not a structured discussion.
  4. In the "Conclusions" part, the conclusion is poor in summarizing the whole paper. Conclusions should be revised.

Round 2

Reviewer 2 Report

I find that the author has made some efforts on this article.I have no another comments.

This manuscript is a resubmission of an earlier submission. The following is a list of the peer review reports and author responses from that submission.

Round 1

Reviewer 1 Report

The muniscipt separately introduced mine tailings and heavy metals’ biological toxicity and their implications on plants, animals, aquatic life and human in detail, but there are several serious problems in this review. The following are the questions in this manuscript:

The text is not well arranged and the logic of needs to be optimized. The title of this review tells the readers that heavy metals from mine tailings is the object of discussion, but the tailings’ source and other source of heavy metals were not clearly destinguished in the contents. Heavy metals in the environment have multiple sources, and the mine tailings source should be the focus of the review. The “introduction” section should emphasize “State-of-the-art”, “research methods”, and “existing problems”of this research field, and the authors should put forward their own opinoins.  Most of the references are very early articles, which cannot reflect the development of this research field.

Reviewer 2 Report

On line 51: The term heavy metal has no standard definition and therefore not very appropriate for usage. The preferred term is toxic metals. Also, the elements As, Se and I have densities which do not meet the threshold of 6.0 g/cm3 set by the writers. The densities for As, Se and I are 5.78, 4.8 and 4,94 g/cm3 respectively.

Line 89: Could the writers provide actual figures to buttress their point?

Line 198: I would suggest that line 198 will be modified to ; the presence of mine tailings in soils lead to acidification.

This is because the  toxic metal ions are normally contained in untreated mine tailings. Low pH tailings, ie acidic tailings, contain higher amount of toxic metals as compared to high pH tailings. At high pH, most of the toxic metal ions form insoluble hydroxides and sulfides which then precipitate to reduce the ion content of the mine tailings.

The article is well organized and the objectives correspond to the conclusions drawn from it.

Reviewer 3 Report

Dear authors

The submitted review seems suitable for their publication in IJERPH. However, I have several considerations about this manuscript that should be reviewed. This review is too generic, and sometimes information is missing (soil organisms such as earthworms) or could be improved.

1. I suggest to review all manuscript and to change the word "heavy metal" by Potentially Toxic Elements or Potentially Hazardous Elements (e.g. line 30-31 or 51-52), due to Al and As aren't heavy metals! (Light metal and metalloid, respectively).

2. Line 54. "Contamination emanating from Heavy metals is usually colourless and odourless" This is not right. Contaminated waters by acid mine drainage can have colour due to the amounts of heavy metals.

3. Line 54-64. I suggest to change this paragraph and to improve their readiness. In the present form, it seems that heavy metals are macronutrients, and Pb is not a nutrient.

4. Sections 2 to 5 are too long and could be joined in the same part about mine tailings because it's too long in the current version. In my opinion, it should be focused how these soils from mining areas are spolic technosols with poorly soil properties (texture, lack of structure, low contents of clay and organic matter, acidic pH and low cation exchange capacity, etc.) due to this information is crucial to understand the behaviour of heavy metals in these areas in terms of bioavailability and risk assessment.

5. Section 7 is too generic. It can be improved with the role of organic matter or iron and manganese oxides on metal mobility and toxicity.

6. Before section 6, it could be interesting to add a section about toxicity to soil organisms such as earthworms or collembola. Soil organisms have vital roles in soil formation and even to biodegradation or metal speciation in mine tailings (worms). In addition to plants, soil organisms are also essential to show.

7. Section 10 (Heavy metals in Human health) it should be focused on the effects on human health related to workers or inhabitants from mine areas. This section is too generic.

Reviewer 4 Report

This study presents a review about mine tailings as a source of heavy metals and its implications on agricultural soils, plants, animals, aquatic life and human health. Due to a widespread of mining activities in South Africa, pollution is a serious health concern with devastating consequences on the health of animals and humans likewise. This paper aims to report contamination of major mining effluents from tailings. In this review, concerted efforts were made to concise the information contained in literature regarding heavy metal pollution and its implications in soil, water, plants, animals, marine life and human health.

Here are some comments about the manuscript:

Article structure is not reasonable, should be divided into several parts according to the hierarchy, the same classification of content into the same category. In the “introduction” part, authors should clearly express the purpose of the research, the significance of this article should be explained. It's not a structured discussion. In the "Conclusions" part, the conclusion is poor in summarizing the whole paper. Conclusions should be revised.